# Short Video Viewing, and Not Sedentary Time, Is Associated with Overweightness/Obesity among Chinese Women

**DOI:** 10.3390/nu14061309

**Published:** 2022-03-21

**Authors:** Ke Chen, Qiang He, Yang Pan, Shuzo Kumagai, Si Chen, Xianliang Zhang

**Affiliations:** 1School of Physical Education, Shandong University, Jinan 250061, China; cchenkee@163.com (K.C.); hq@sdu.edu.cn (Q.H.); panyang@sdu.edu.cn (Y.P.); 2Center for Health Science and Counseling, Kyushu University, Fukuoka 819-0395, Japan; kumagai.shuzo.296@m.kyushu-u.ac.jp; 3School of Nursing and Rehabilitation, Cheeloo College of Medicine, Shandong University, Jinan 250012, China

**Keywords:** short videos, sedentary time, older women, overweightness/obesity, tri-axial accelerometer

## Abstract

Previous studies have found that the relationship between sedentary time (ST) and overweightness/obesity is unclear. The association between sedentary behavior and overweightness/obesity may depend on the type of sedentary behavior engaged in. Nowadays, in older Chinese adults, especially females, short video viewing (SVV) is the most popular leisure sedentary behavior. However, the association between SVV and overweightness/obesity remains to be determined. This study aimed to examine the associations between ST and SVV and overweightness/obesity in Chinese community-dwelling older women. A cross-sectional analysis of baseline data from the Physical Activity and Health in Older Women Study was carried out in this study. A total of 1105 older Chinese women aged 60–70 years were included. SVV was estimated using a self-reported questionnaire, and ST was objectively measured using a tri-axial accelerometer. Overweightness/obesity indicators, including body fat ratio (BFR), fat mass (FM), visceral fat mass (VFM), subcutaneous fat mass (SFM), trunk fat mass (TFM), and limb fat mass (LFM), were assessed using multi-frequency bioimpedance analysis. The covariates included socio-demographic data and a range of health-related factors. Multiple linear regression analyses were used to assess the association between ST and SVV and overweightness/obesity. ST was significantly positively associated with all indicators of overweightness/obesity; however, the associations disappeared after adjusting for moderate-to-vigorous-intensity physical activity (MVPA). A higher SVV time was associated with a higher body mass index (BMI) (β = 0.19, 95% confidence interval (CI): 0.05 to 0.32), BFR (β = 0.31, 95% CI: 0.07 to 0.56), FM (β = 0.33, 95% CI: 0.04 to 0.61), VFM (β = 0.09, 95% CI: 0.01 to 0.16), SFM (β = 0.24, 95% CI: 0.03 to 0.45), TFM (β = 0.21, 95% CI: 0.04 to 0.39), and LFM (β = 0.11, 95% CI: 0.00 to 0.23) in the fully adjusted models. Compared with non-food short videos, short food videos had a greater effect on overweightness/obesity. SVV was an independent risk factor for overweightness/obesity. A reduction in SVV (especially the food category) rather than ST might be an effective way to prevent overweightness/obesity when incorporated in future public health policy formulations.

## 1. Introduction

The number of overweight/obese older adults has been steadily increasing, and the prevalence of central obesity is the highest among older women (60–69 years) [1,2]. Overweightness/obesity is associated with an increased risk of cardiovascular diseases, such as hypertension and heart failure, atrial fibrillation, and stroke [3,4]. Past studies have shown that sedentary behavior is an independent risk factor for overweightness/obesity [5,6]. Reducing sedentary time (ST) in older adults seems to be a promising strategy for preventing overweightness/obesity [7]. However, accumulating findings concerning the relationship between ST and overweightness/obesity are controversial [8,9,10]. In a review of reviews, Biddle et al. [11] considered that the effects of sedentary behavior on overweightness/obesity might depend on the types of sedentary behavior engaged in, such as television viewing (TV), which was more strongly associated with overweightness/obesity. This may be because people who often watch TV may watch more advertisements and eat more high-calorie snacks. When energy intake continues to exceed energy consumption, it will lead to metabolic diseases and obesity [12]. It should be noted that previous studies on this topic have relied on self-reported overweightness/obesity indicators [13,14,15] and ignored whether the association between TV and overweightness/obesity was independent of moderate-to-vigorous physical activity (MVPA) [16,17,18].

With the development of information technology, short video viewing (SVV) has gradually replaced TV as one of the most prevalent sedentary behaviors in the world. According to the latest data, by December 2020, the number of Internet users in China had reached 989 million, of which 873 million watched short videos, accounting for 88.3% of Internet users [19]. Older women account for a high percentage of SVV and are more willing to view short videos than older men [20]. So far, there has been little discussion about the relationship between SVV and ST and overweightness/obesity in older women. In addition to SVV time, the SVV content may also be associated with overweightness/obesity. A recent study found that exposure to social media food videos influenced people’s liking of the foods and their food choice behavior [21]. Since short video platforms can accurately post advertisements to target groups, food marketing has shifted food exposure choices toward short video platforms. Older Chinese women still traditionally fulfill the role of homemaker and undertake the tasks of cooking and buying food; however, very few studies have focused on the relationship between SVV content and overweightness/obesity in this population [22,23].

Therefore, the objectives of the current study were to investigate the associations between SVV time and ST and overweightness/obesity and to analyze the association between SVV content (food and non-food) and overweightness/obesity in Chinese community-dwelling older women.

## 2. Materials and Methods

### 2.1. Participants and Study Design

Data were obtained from the baseline survey (March to June 2021) of the Physical Activity and Health in Older Women Study (PAHIOWS). The PAHIOWS is an ongoing community-based cohort study in Yantai City, Shandong Province, China, aiming to explore the associations between physical activity, sedentary behavior, and health outcomes among older women. Participants for the study were recruited from a community center. The inclusion criteria were as follows: community-dwelling women aged 60–70 years, able to communicate, with no cognitive impairment (according to the Mini Mental State Examination (MMSE) score: >17 for illiteracy, >20 for primary school, and >24 for middle school and above), and a willingness to provide informed consent. Face-to-face interviews with participants, including a description of the study, the informed consent document, a questionnaire, and the instructions for using and wearing the accelerometer, were performed by trained staff. The participants’ anthropometric data and overweightness/obesity indicators were measured. A total of 1370 women participated in the present study; 195 individuals who provided less than 4 days of accelerometer data, 34 individuals with invalid overweightness/obesity indicator data, and 36 individuals who had missing values for covariates were excluded. After exclusion, 1105 participants were included in the final analysis. The Institutional Review Board of the School of Nursing and Rehabilitation, Shandong University (2020-R-001) approved the study protocol, and all participants provided written informed consent before participating in the study.

### 2.2. SVV and ST

SVV was measured using a questionnaire. The respondents were asked: (1) “Did you view short videos, yes or no?” (2) “How many hours or minutes did you view short videos on an average day?” (3) “Did you view short food videos, yes or no?”.

Actigraph wGT3X-BT accelerometers (ActiGraph, Pensacola, FL, USA) were used to estimate the body movement intensity and duration. Participants were asked to wear the accelerometer on the left side of the waist, which was secured via a belt. They were instructed to put the accelerometer on when they awoke and take it off only before going to bed or when participating in water activities. The research staff contacted participants twice via telephone to ensure that the accelerometers were worn correctly. Once the 7-day wearing period was complete, according to the manufacturer’s specifications, data from the accelerometer were downloaded to a personal computer and subsequently imported to ActiLife software (Version 6.13.4, ActiGraph, Pensacola, FL, USA), where data could be examined minute-by-minute. Each accelerometer file was subjected to the following standardized data quality procedures to assess the reliability and validity of the data. Non-wear time was defined as ≥90 consecutive minutes of no activity, and these data were removed from the analysis. The data during which the accelerometer was worn for ≥4 valid wear days (≥10 h of wear time/day) were included in the analysis [24]. Accelerometer counts, during the time worn, were classified as ≤99 counts/min for ST, and ≥2020 counts/min for MVPA [25]. Total daily minutes of sedentary activity and MVPA were summed and averaged across all valid days.

### 2.3. Overweightness/Obesity Measures

Body mass index (BMI) was calculated from the body height and weight, which were measured to the nearest 0.1 cm and 0.1 kg, respectively. Assessment of overweightness/obesity indicators was performed by multi-frequency bioimpedance analysis using the Tanita MC-180 (Bailida Co., Tokyo, Japan). This device is considered appropriate for evaluating overweightness/obesity indicators in the target population because its validity for assessing total and visceral adiposity in older women has been confirmed [26,27]. Overweightness/obesity indicators were measured using standard protocols with the subject in minimal clothes and without shoes on the instrument. The height-adjusted BMI, body fat ratio (BFR), fat mass (FM), visceral fat mass (VFM), subcutaneous fat mass (SFM), trunk fat mass (TFM), and limb fat mass (LFM) were used as indicators of obesity.

### 2.4. Covariate Variables

Socio-demographic variables (age, income, and living alone) and health-related factors (alcohol intake, chronic conditions, insomnia, health-related quality of life, nutritional assessment, anxiety, depression, and cognitive function) were recorded in the questionnaires. Individual monthly income (≤¥1000, ¥1001–2000, ¥2001–3000, ¥3001–4000, >¥4000) and living alone (yes or no) were collected for each participant. General healthy-living indicators included self-reported alcohol intake: (a) non-consumers, with no alcohol intake in the past year, and (b) alcohol consumers, with alcohol intake more than once a year; and chronic conditions (yes or no), where ‘yes’ represents whether the participant suffers from at least one chronic disease and ‘no’ where the participant does not suffer from any chronic diseases. Moreover, the Chinese version of the Athens Insomnia Scale (AIS), which consists of eight measurement items of sleep features, was used to assess sleep; the higher the score, the more severe insomnia complaints were in general. This scale has satisfactory internal consistency (0.83) and test-retest reliability (0.94) [28].

Health-related quality of life was measured using the European Quality of life 5-dimensions (EQ-5D) and the visual analog scale (EQ-VAS) according to the parameters of the Chinese population. A higher EQ-5D and EQ-VAS score means a better quality of life. These scales have been widely used in the study of Chinese people and have acceptable reliability and validity [29,30]. The 2-item Generalized Anxiety Disorder Scale (GAD-2) and the 2-item Patient Health Questionnaire (PHQ-2) were used to assess anxiety and depression. GAD-2 and PHQ-2 were used as continuous scales, where the higher the score, the greater the anxiety and depression. The validity of GAD-2 (area under the curve, 0.80 to 0.91) [31] and PHQ-2 (sensitivity of 83% and a specificity of 92%) [32] has been demonstrated. Nutritional assessment was measured using the short-form mini-nutritional assessment (MNA-SF) and categorized as: normal nutrition status (12–14 score), at risk of malnutrition (8–11 score), or malnourished (0–7 score) [33]. For MNA-SF (cutoff point ≤ 11), average sensitivities and specificities were 0.95 and 0.95 [34]. Cognitive function was assessed using the MMSE score (range 0–30 score); a lower score indicated poorer cognitive function [35]. The optimal threshold of the original MMSE screening produced 72.5% and 91.3% sensitivity and specificity estimates [36].

### 2.5. Statistical Analyses

Participant characteristics are presented as a median (interquartile range, IQR) for continuous variables and aa number (%) for categorical variables. The chi-square test and Kruskal–Wallis of variance were used to assess the differences between the groups. Multiple linear regression models were used to estimate β coefficients with 95% confidence intervals (CIs) for the associations between each indicator of overweightness/obesity and ST and SVV. The following 2 models were used to adjust for confounding factors: Model 1 included age plus wear time, income, living alone, drinking, disease, EQ-5D score, EQ-VAS score, GAD-2 score, AIS score, PHQ-2 score, MNA-SF, and MMSE score, and for SVV, ST was also added to Model 1. Model 2 included factors in Model 1 plus the total MVPA time. The coefficient of variance expansion (VIF) for all variables was calculated to detect the existence of collinearity. In the fully adjusted model, each covariate had a VIF below 5, which is considered acceptable. In addition, we defined a one-unit increment in ST as one hour. The analysis was performed using Stata SE, version 16 (Stata Corp., College Station, TX, USA). Prism8 (GraphPad Software Inc., San Diego, CA, USA) was used to generate plots.

## 3. Results

The socio-demographic characteristics, overweightness/obesity indicators, physical activity, and sedentary behavior of the participants are shown in Table 1. All community-dwelling women were 65 (IQR 63.00–67.00) years of age, 10.95% lived alone, 34.84% had no chronic disease, 9.50% drank alcohol, and they had a BMI of 25.20 (IQR 23.20–27.40) kg/m^2^. The women spent 8.81 (IQR 7.87–9.98) h/day in ST and 0.47 (IQR 0.27–0.68) h/day in MVPA. Significant differences were observed between non-SVV, viewing non-food short videos, and short food videos in terms of age, overweightness/obesity indicators, and MVPA. However, no differences in income, living situation, chronic disease, drinking, ST, and wear time were observed between the groups.

Figure 1 displays the results of the association between ST and the indicators of overweightness/obesity. Significant associations between ST and BFR, FM, VFM, SFM, TFM, and LFM were observed in Model 1 while all these associations disappeared after adjusting for MVPA.

Figure 2 displays the results of the association between SVV time and indicators of overweightness/obesity. Significant associations between ST and BMI, BFR, FM, VFM, SFM, TFM, and LFM were observed in all models, indicating that engagement in MVPA cannot counteract the negative influence of SVV time.

Figure 3 displays the results of the association between SVV content and overweightness/obesity. Significant associations between SVV content and BMI, BFR, FM, VFM, SFM, TFM, and LFM were found in all models, and the coefficients of short food videos with overweightness/obesity indicators were larger than those of non-food short videos.

## 4. Discussion

To the best of our knowledge, this is the first study to report an association between SVV and overweightness/obesity. SVV time and SVV content were independent risk factors for overweightness/obesity. Compared to non-food short videos, viewing short food videos was more relevant. ST was significantly and positively correlated with overweightness/obesity; however, this impact vanished after adjusting for MVPA.

Our results showed that ST was associated with overweightness/obesity but not with MVPA. In accordance with the present results, Wanner [37] and Foong et al. [38] demonstrated that MVPA could offset the relationship between ST and overweightness/obesity. This finding was also echoed by previous evidence, where Peterson et al. [39] found that significant elevations in obesity, a reduction in high-density lipoprotein cholesterol, and increases in total cholesterol and triglycerides were linked to less MVPA, regardless of ST. Similarly, Ekelund et al. [40] reported that the relationship between ST and mortality rates is weaker when levels of MVPA are attained for 30–40 min. However, the results reported by other authors, such as Swartz et al. [41,42,43], were not inconsistent with our own, and the reasons for the inconsistency might be as follows: (1) The statistical methods were divergent. Swartz et al. [41] used Pearson’s correlation to assess the correlation between overweightness/obesity and ST while in the present study, we used multiple linear regression to control for multiple confounders. (2) The measures of overweightness/obesity were different. Stamatakis et al. [42] used anthropometry to characterize overweightness/obesity, whereas, in our study, we used multi-frequency bioimpedance, which was more accurate. (3) Different types of accelerometers were used. Zhu et al. [43] used a uniaxial accelerometer to measure physical activity while this study used a triaxial accelerometer, which has a higher association with energy expenditure. (4) The control for confounding variables was incongruent. Du et al. [44] did not control for physical activity while this study controlled for objective MVPA time.

Unlike ST, there was a significant positive relationship between SVV time and overweightness/obesity after adjusting for MVPA, ST, and other potential confounders. There are several plausible explanations for these results. First, women who receive negative information from social media are more likely to perceive stress [45]. Stress, in turn, promotes cortisol production by activating the hypothalamic-pituitary-adrenal axis; cortisol can directly promote abdominal fat deposition [46]. Secondly, short video apps can predict each user’s interest and recommend personalized videos. This can activate the default mode network, reduce cognitive control, and develop a problematic usage pattern that manifests as addictive undesirable behavior [47]. This further leads to increased screen time, ultimately reducing melatonin production. The decrease in melatonin may be related to diurnal rhythms of the gut microbiota, in turn affecting lipid metabolism [48]. Previous cross-sectional studies discussing the relationship between TV time and overweightness did not consider TV content. However, in the present study, we also found that SVV content was associated with overweightness/obesity, independent of MVPA. This is not surprising, as a previous study found that frequent exposure to short food videos can increase the desire for food, leading to increased eating, resulting in overweightness/obesity [49]. Exposure to unhealthy dietary advertising promotes poor diets, and excess energy intake has also been previously reported in the literature [50,51].

In recent years, increasing attention has been paid to the benefits of small changes to behaviors in public health. Identifying the determinants of overweightness/obesity, especially those that are easily modifiable, is necessary to develop effective interventions targeted at reducing overweightness/obesity. A meta-analysis involving 1 million people showed that 60–75 min per day of moderate physical activity appears to eliminate the increased mortality risk associated with ST. Nevertheless, MVPA cannot eliminate the increased risk associated with TV time [52]. This view seems to be consistent with our research, which found that changing the types of sedentary behavior may be more important than reducing ST for weight loss. Based on this evidence, one could speculate that a promising strategy for overweightness/obesity prevention might be to reduce SVV and engage in other activities, given the increasing prevalence of SVV, the accelerated aging of the population, and the broad range of overweightness/obesity. Reduction of SVV may hence be a more effective measure for public health interventions than the previously recognized measures.

Several limitations need to be considered in this study. First, this study was cross-sectional, carrying with it the limitations of a cross-sectional design, and therefore it could not determine causality. In the future, more randomized control trials and longitudinal studies are required to validate our results. Another limitation was that all the participants in our study were from Yantai City, Shandong Province. These findings may not be able to be extrapolated to the entire population of China. Finally, participation was based on self-selection; those who were willing to participate in this study might have been healthier people.

## 5. Conclusions

This is the first community-based cross-sectional study to explore the association between SVV time and SVV content and overweightness/obesity. The results revealed that SVV time and SVV content, instead of ST, were independent risk factors for overweightness/obesity in the elderly female population. Our findings complement those studies using cross-sectional designs relating to screen time and extend them to older adults, which can be used to formulate policies to prevent short video addiction.

## Figures and Tables

**Figure 1 nutrients-14-01309-f001:**
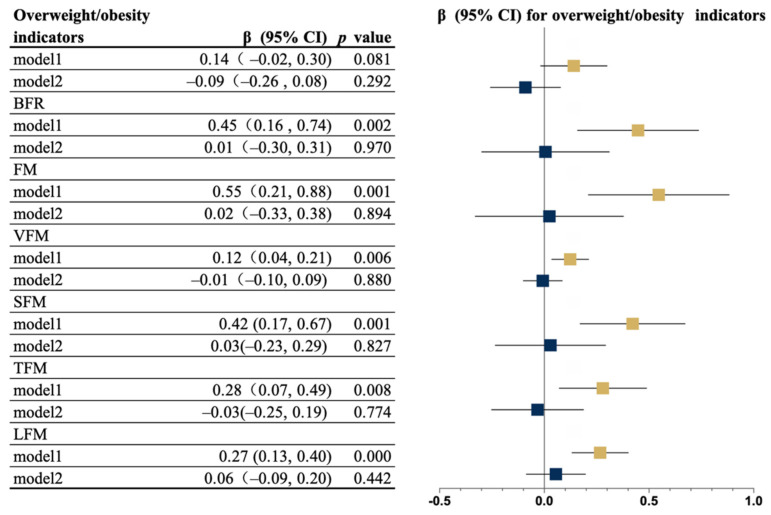
Multivariable associations between ST and overweightness/obesity indicators (*n* = 1105). Model 1: Adjusted for age, income, living alone, drinking, disease, EQ-5D score, EQ-VAS score, GAD-2 score, AIS score, PHQ-2 score, MNA-SF, MMSE score, and wear time; Model 2: Additional adjustment for MVPA time. Yellow and blue indicate Models 1 and 2, respectively. The time is represented by squares. Error bars represent 95% confidence intervals (CIs). BMI, body mass index; BFR, body fat ratio; FM, fat mass; LFM, limb fat mass; SFM, subcutaneous fat area; ST, sedentary time; TFM, trunk fat mass; and VFM, visceral fat mass.

**Figure 2 nutrients-14-01309-f002:**
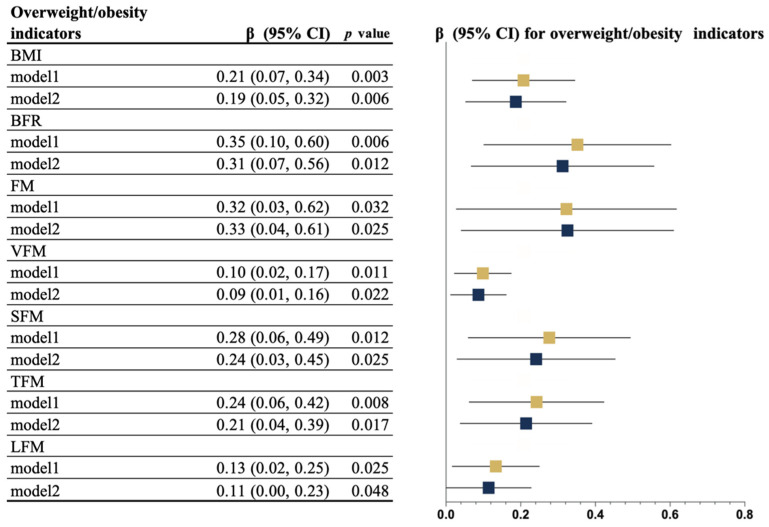
Multivariable associations between SVV time and overweightness/obesity indicators (*n* = 1105). Model 1: Adjusted for age, income, living alone, drinking, disease, EQ-5D score, EQ-VAS score, GAD-2 score, AIS score, PHQ-2 score, MNA-SF, MMSE score, ST, and wear time; Model 2: Additional adjustment for MVPA time. Yellow and blue indicate Models 1 and 2, respectively. The time is represented by squares. Error bars represent 95% confidence intervals (CIs). BMI, body mass index; BFR, body fat ratio; FM, fat mass; LFM, limb fat mass; SFM, subcutaneous fat area; SVV, short video viewing; TFM, trunk fat mass; and VFM, visceral fat mass.

**Figure 3 nutrients-14-01309-f003:**
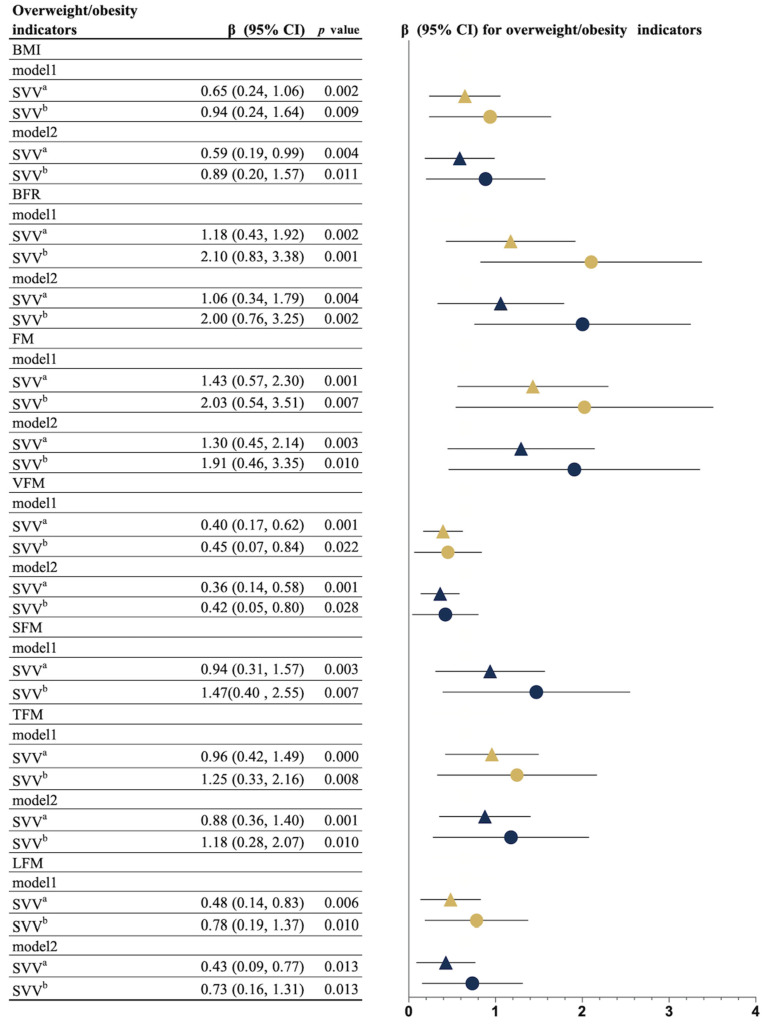
Multivariable associations between SVV content and overweightness/obesity indicators (*n* = 1105). ^a^ non-food category; ^b^ food category. Model 1: Adjusted for age, income, living alone, drinking, disease, EQ-5D score, EQ-VAS score, GAD-2 score, AIS score, PHQ-2 score, MNA-SF, MMSE score, ST, and wear time; Model 2: Additional adjustment for MVPA time. Yellow and blue indicate models 1 and 2, respectively. Non-food short videos are represented by triangles, and short food videos are represented as circles. Error bars represent 95% confidence intervals (CIs). BMI, body mass index; BFR, body fat ratio; FM, fat mass; LFM, limb fat mass; SFM, subcutaneous fat area; SVV, short video viewing; TFM, trunk fat mass; and VFM, visceral fat mass.

**Table 1 nutrients-14-01309-t001:** Characteristics of the study population (*n* = 1105).

		Daily Exposure			
Characteristics	Total	Non–SVV (*n* = 370)	SVV ^a^ (*n* = 636)	SVV ^b^ (*n* = 99)	*p* Value
Age (years)	65.00 (63.00–67.00)	65.00 (63–68)	65.00 (63.00–67.00)	64.00 (62.00–66.00)	0.020
Income, (¥/month)					
≤¥1000	35 (3.17)	14 (40.00)	17 (48.57)	4 (11.43)	
¥1001–2000	102 (9.23)	36 (35.29)	64 (62.75)	2 (1.96)	
¥2001–3000	181 (16.38)	58 (32.04)	106 (58.56)	17 (9.39)	0.303
¥3001–4000	565 (51.13)	182 (32.21)	330 (58.4)	53 (9.38)	
>¥4000	222 (20.09)	80 (36.04)	119 (53.60)	23 (10.36)	
Living alone	121 (10.95)	46 (38.02)	69 (57.02)	4 (4.96)	0.195
No chronic disease	385 (34.84)	123 (31.95)	232 (60.26)	30 (7.79)	0.356
Current drinker	105 (9.50)	32 (30.48)	65 (61.90)	8 (7.62)	0.629
Overweightness/obesity Indicators					
BMI (kg/m^2^)	25.20 (23.20–27.40)	24.70 (22.90–26.80)	25.40 (23.40–27.60)	25.70 (23.70–28.20)	0.007
BFR (%)	35.90 (32.10–39.80)	35.25 (31.50–39.20)	36.20 (32.40–40.25)	37.40 (33.80–40.20)	0.003
FM (kg)	22.90 (18.90–27.60)	22.05 (18.30–26.50)	23.05 (19.20–28.10)	24.40 (20.20–27.80)	0.003
VFM (kg)	3.60 (2.80–4.90)	3.45 (2.60–4.70)	3.70 (2.80–5.10)	4.00 (3.00–5.10)	0.004
SFM (kg)	19.20 (16.30–22.70)	18.70 (15.60–21.90)	19.40 (16.40–23.05)	20.40 (17.00–23.10)	0.002
TFM (kg)	13.30 (10.70–16.30)	12.60 (10.30–15.60)	13.40 (10.90–16.50)	13.90 (11.60–16.90)	0.003
LFM (kg)	9.80 (8.30–11.60)	9.55 (8.10–11.20)	9.80 (8.40–11.70)	10.50 (8.80–12.10)	0.003
Physical activity and sedentary behavior					
ST (h/day)	8.81 (7.87–9.98)	8.73 (7.75–9.97)	8.87 (7.97–10.02)	8.81 (7.93–9.76)	0.389
MVPA (h/day)	0.47 (0.27–0.68)	0.49 (0.30–0.71)	0.45 (0.25–0.66)	0.48 (0.26–0.71)	0.039
wear time (h/day)	14.50 (13.48–15.62)	14.57 (13.59–15.77)	14.46 (13.41–15.58)	14.35 (13.54–15.10)	0.189

^a^ non-food category; ^b^ food category. Continuous variables are reported as the median with an interquartile range, and categorical variables are reported as a number with a percentage. Based on chi square test (categorical variables) or Kruskal–Wallis (continuous variables) Abbreviations: BMI, body mass index; BFR, body fat ratio; CNY, Chinese yuan; FM, fat mass; LFM, limb fat mass; MVPA, moderate-to-vigorous physical activity; SFM, subcutaneous fat area; ST, sedentary time; SVV, short video viewing; TFM, trunk fat mass; VFM, visceral fat mass.

## Data Availability

The data presented in this study are available upon reasonable request from the corresponding author.

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
