# Peer review of "Short Video Viewing, and Not Sedentary Time, Is Associated with Overweightness/Obesity among Chinese Women"

_nutrients, 2022, doi:10.3390/nu14061309_

Round 1

Reviewer 1 Report

Dear author,

Is an honour to have the possibility to evaluated your manuscript. Is interesting to analysed the association with overweight/obesity and short video viewing in Chinese older woman.

Statistical Analysis

Normality test was used to analysis the distribution of the variables? (Shapiro wilk or Kolmogorov Smirnov) should be described and added if one of them had not normal distribution, which statistical test was used.

Results.

(Table 1) At the bottom of the table should be described the statistical test used, to stablish the signification.

Best regards,

Author Response

Dear Editors and Reviewers,

Thanks very much for taking your time to review this manuscript. We really appreciate all your generous comments and suggestions! According to your advice, we have carefully considered and corrected the manuscript for revision. The revision instructions are as follows:

Point 1: Normality test was used to analysis the distribution of the variables? (Shapiro wilk or Kolmogorov Smirnov) should be described and added if one of them had not normal distribution, which statistical test was used.

Response 1: Thank you very much for your valuable time and energy. We thank you again for your work. Before processing the data, all of the data were tested for normal distribution. Data were deemed approximately normally distributed as determined by Q-Q plots and histogram. However, using Shapiro Wilk or Kolmogorov Smirnov to test, it was found that the P values were less than 0.05. Then we use the nonparametric (Kruskal Wallis test) to test, which is consistent with the previous conclusion. We now attach the results of the nonparametric test here. The statistical method has been marked at the bottom of Figure 1 according to your suggestions.

                        Table 1. Characteristics of the study population (n = 1,105).

Daily exposure

Characteristics

Total

Non-SVV

SVVa

SVVb

Pvalue

(n=370)

(n=636)

(n=99)

Age (years)

65.00 (63.00-67.00)

65.00(63-68)

65.00 (63.00-67.00)

64.00 (62.00-66.00)

0.020

Income, (¥/month)

≤¥1000

35 (3.17)

14 (40.00)

17 (48.57)

4 (11.43)

¥1001-2000

102 (9.23)

36 (35.29)

64 (62.75)

2 (1.96)

¥2001-3000

181 (16.38)

58 (32.04)

106 (58.56)

17 (9.39)

0.303

¥3001-4000

565 (51.13)

182 (32.21)

330 (58.4)

53 (9.38)

>¥4000

222 (20.09)

80 (36.04)

119 (53.60)

23 (10.36)

Living alone

121 (10.95)

46 (38.02)

69 (57.02)

4 (4.96)

0.195

No chronic disease

385 (34.84)

123 (31.95)

232 (60.26)

30 (7.79)

0.356

Current drinker

105 (9.50)

32 (30.48)

65 (61.90)

8 (7.62)

0.629

      Overweight/obesity Indicators

BMI (kg/m2)

25.20 (23.20-27.40)

24.70 (22.90-26.80)

25.40 (23.40-27.60)

25.70 (23.70-28.20)

0.007

BFR (%)

35.90 (32.10-39.80)

35.25 (31.50-39.20)

36.20 (32.40-40.25)

37.40 (33.80-40.20)

0.003

FM (kg)

22.90 (18.90-27.60)

22.05 (18.30-26.50)

23.05 (19.20-28.10)

24.40 (20.20-27.80)

0.003

VFM (kg)

3.60 (2.80-4.90)

3.45 (2.60-4.70)

 3.70 (2.80-5.10)

4.00 (3.00-5.10)

0.004

SFM (kg)

19.20 (16.30-22.70)

18.70 (15.60-21.90)

19.40 (16.40-23.05)

20.40 (17.00-23.10)

0.002

TFM (kg)

13.30 (10.70-16.30)

12.60 (10.30-15.60)

13.40 (10.90-16.50)

13.90 (11.60-16.90)

0.003

LFM (kg)

9.80 (8.30-11.60)

9.55 (8.10-11.20)

9.80 (8.40-11.70)

10.50 (8.80 -12.10)

0.003

       Physical activity and sedentary behavior

ST (h/d)

8.81 (7.87-9.98)

8.73 (7.75-9.97)

8.87 (7.97-10.02)

8.81 (7.93-9.76)

0.389

MVPA (h/d)

0.47 (0.27-0.68)

0.49 (0.30-0.71)

0.45 (0.25-0.66)

0.48 (0.26-0.71)

0.039

wear time (h/d)

14.50 (13.48 -15.62)

14.57 (13.59-15.77)

14.46 (13.41-15.58)

14.35 (13.54-15.10)

0.189

a non-food category; b food category. Continuous variables are reported as median with interquartile range, and categorical variables are reported as number with percentage. Based on chi square test (categorical variables) or Kruskal-Wallis (continuous variables). Abbreviations: BMI, body mass index; BFR, body fat ratio; CNY, Chinese yuan; FM, fat mass; LFM, limb fat mass; MVPA, moderate-to-vigorous physical activity; SFM, subcutaneous fat area; ST, sedentary time; SVV, short video viewing; TFM, trunk fat mass; VFM, visceral fat mass.

Thank again for your hard work and valuable advice.

Author: Ke Chen

16 Mar 2022

Reviewer 2 Report

Comments to the work titled “Short video viewing, not sedentary time, is associated with overweight/obesity among Chinese older women”

1.It seems that other (apart from low physical activity) causes of excessive body mass (important role of the nutritional factor) could also be mentioned in the ‘Introduction’ section.
2. Was the questionnaire used to assess viewing short videos subject to a validation procedure or was its reliability assessed?
3. It appears that the implemented tools can be described in more detail, given in section 2.4.
4.Table 1, for the indices of obesity and physical activity, there is no information provided on the type of descriptive statistics indicated (mean and SD) (while such info was given for other variables in the table).

From a linguistic point of view, the text requires minor revisions.
For example, the title of the work should be changed from “Short video viewing, not sedentary time, is associated with overweight/obesity among Chinese older women” to “Short Video Viewing, and not Sedentary Time, is Associated with Overweightness/Obesity Among Older Chinese Women”.
Alterations should regard usage of the passive voice, correcting article usage, punctuation, word order, eradication of repetitions, etc. Please, consult a native speaker.

Author Response

Dear Editors and Reviewers,

Thanks very much for taking your time to review this manuscript. We really appreciate all your generous comments and suggestions! According to your advice, we have carefully considered and corrected the manuscript for revision. The revision instructions are as follows:

Point 1: It seems that other (apart from low physical activity) causes of excessive body mass (important role of the nutritional factor) could also be mentioned in the ‘Introduction’ section.

Response 1: Thank you very much for your suggestion. It is very valuable to improve the content of our article. It has been added in the introduction. The added content is: This may be because people who often watch TV may watch more advertisements and eat more high calorie snacks. When energy intake continues to exceed energy consumption, it will lead to metabolic diseases and obesity[12].

Point 2: Was the questionnaire used to assess viewing short videos subject to a validation procedure or was its reliability assessed?

Response 2: We appreciate your question. In addition, in order to ensure the accuracy of the questionnaire, we use one-to-one, face-to-face inquiry, and all personnel have been specially trained. Our questionnaire is similar to the previous questionnaire on TV viewing time [1], and face validity was established by experts. Moreover, according to the questionnaire survey conducted from March to July 2021, the average length of watching short videos per day is 1.74 ± 1.37h, which is similar to the result of per capita daily short video use time (125min) in the statistical report on the development of China's Internet Network released by China Internet Center in September 2021 [2].

  1. Hoang TD, Reis J, Zhu N, et al. Effect of early adult patterns of physical activity and television viewing on midlife cognitive function. JAMA Psychiatry 2016;73:73-9.
  2. China internet network information center. The 48th Statistical Report on Internet Development in China. Available online: http://www.cnnic.cn/hlwfzyj/hlwxzbg/

Point 3: It appears that the implemented tools can be described in more detail, given in section 2.4.

Response 3: Thank you for your suggestion. We have made changes in Section 2.4 to add the classification variables or continuous variables in detail, and explain the score of the scale. The amendments are as follows:

Individual monthly income (≤¥1000, ¥1001-2000, ¥2001-3000, ¥3001-4000, >¥4000) and living alone (yes or no ) were collected for each participant. General healthy living indicators included self-reported alcohol intake: (a) non-consumers, with no alcohol intake in the past year, and (b) alcohol consumers, with alcohol intake more than once a year, and chronic conditions (yes or no), where ‘yes’ represents whether the participant suffers from at least one chronic disease and ‘no’ where the participant does not suffer from any chronic diseases. Moreover, the Chinese version of the Athens Insomnia Scale (AIS), which consists of eight measurement items of sleep features, was used to assess sleep; the higher the score, the more severe insomnia complaints were in general.This scale has satisfactory internal consistency (0.83) and test-retest reliability (0.94) [28].

Health-related quality of life was measured using the European Quality of life 5-dimensions (EQ-5D) and the visual analog scale (EQ-VAS) according to the parameters of the Chinese population. Higher EQ-5D and EQ-VAS score means a better quality of life. These scales have been widely used in the study of Chinese people and have acceptable reliability and validity [29,30]. The 2-item Generalized Anxiety Disorder Scale (GAD-2) and the 2-item Patient Health Questionnaire (PHQ-2) were used to assess anxiety and depression. GAD-2 and PHQ-2 were used as continuous scales, the higher the score, the greater the anxiety and depression. The validity of GAD-2 (area under the curve, 0.80 to 0.91) [31] and PHQ-2 ( sensitivity of 83% and a specificity of 92% ) [32] has been demonstrated. Nutritional assessment was measured using the short-form mini-nutritional assessment (MNA-SF) and categorized as: normal nutrition status (12-14 score), at risk of malnutrition (8-11 score), or malnourished (0-7 score) [33]. For MNA-SF (cutoff point ≤11), average sensitivities and specificities were 0.95 and 0.95 [34]. Cognitive function was assessed using the MMSE score(range 0–30 score); a lower score indicated poorer cognitive function [35]. The optimal threshold of the original MMSE screening produced 72.5% and 91.3% sensitivity and specificity estimates [36].

Point 4: Table 1, for the indices of obesity and physical activity, there is no information provided on the type of descriptive statistics indicated (mean and SD) (while such info was given for other variables in the table).

Response 4: Thank you very much for your reminder. We have made changes in Table 1. Continuous variables are reported as median, and classified variables are reported as a percentage,and marked below table 1.

Table 1. Characteristics of the study population (n = 1,105).

Daily exposure

Characteristics

Total

Non–SVV

SVVa

SVVb

Pvalue

(n=370)

(n=636)

(n=99)

Age (years)

65.00 (63.00–67.00)

65.00(63–68)

65.00 (63.00–67.00)

64.00 (62.00–66.00)

0.020

Income, (¥/month)

≤¥1000

35 (3.17)

14 (40.00)

17 (48.57)

4 (11.43)

¥1001–2000

102 (9.23)

36 (35.29)

64 (62.75)

2 (1.96)

¥2001–3000

181 (16.38)

58 (32.04)

106 (58.56)

17 (9.39)

0.303

¥3001–4000

565 (51.13)

182 (32.21)

330 (58.4)

53 (9.38)

>¥4000

222 (20.09)

80 (36.04)

119 (53.60)

23 (10.36)

Living alone

121 (10.95)

46 (38.02)

69 (57.02)

4 (4.96)

0.195

No chronic disease

385 (34.84)

123 (31.95)

232 (60.26)

30 (7.79)

0.356

Current drinker

105 (9.50)

32 (30.48)

65 (61.90)

8 (7.62)

0.629

Overweightness/obesity Indicators

BMI (kg/m2)

25.20 (23.20–27.40)

24.70 (22.90–26.80)

25.40 (23.40–27.60)

25.70 (23.70–28.20)

0.007

BFR (%)

35.90 (32.10–39.80)

35.25 (31.50–39.20)

36.20 (32.40–40.25)

37.40 (33.80–40.20)

0.003

FM (kg)

22.90 (18.90–27.60)

22.05 (18.30–26.50)

23.05 (19.20–28.10)

24.40 (20.20–27.80)

0.003

VFM (kg)

3.60 (2.80–4.90)

3.45 (2.60–4.70)

3.70 (2.80–5.10)

4.00 (3.00–5.10)

0.004

SFM (kg)

19.20 (16.30–22.70)

18.70 (15.60–21.90)

19.40 (16.40–23.05)

20.40 (17.00–23.10)

0.002

TFM (kg)

13.30 (10.70–16.30)

12.60 (10.30–15.60)

13.40 (10.90–16.50)

13.90 (11.60–16.90)

0.003

LFM (kg)

9.80 (8.30–11.60)

9.55 (8.10–11.20)

9.80 (8.40–11.70)

10.50 (8.80–12.10)

0.003

Physical activity and sedentary behavior

ST (h/d)

8.81 (7.87–9.98)

8.73 (7.75–9.97)

8.87 (7.97–10.02)

8.81 (7.93–9.76)

0.389

MVPA (h/d)

0.47 (0.27–0.68)

0.49 (0.30–0.71)

0.45 (0.25–0.66)

0.48 (0.26–0.71)

0.039

wear time (h/d)

14.50 (13.48 –15.62)

14.57 (13.59–15.77)

14.46 (13.41–15.58)

14.35 (13.54–15.10)

0.189

a non-food category; b food category. Continuous variables are reported as median with interquartile range, and categorical variables are reported as number with percentage. Based on chi square test (categorical variables) or Kruskal-Wallis (continuous variables) Abbreviations: BMI, body mass index; BFR, body fat ratio; CNY, Chinese yuan; FM, fat mass; LFM, limb fat mass; MVPA, moderate-to-vigorous physical activity; SFM, subcutaneous fat area; ST, sedentary time; SVV, short video viewing; TFM, trunk fat mass; VFM, visceral fat mass.

In addition, we have revised the title according to your suggestion, and the manuscript has been polished by MDPI,English editing invoice: english-41585

Thank again for your hard work and valuable advice.

Author: Ke Chen

16 Mar 2022
